# Just Avoid Robust Inaccuracy: Boosting Robustness Without Sacrificing Accuracy

**Yannick Merkli**[1]    **Pavol Bielik**[2]    **Petar Tsankov**[2]    **Martin Vechev**[1]

[1]ETH Zurich, Switzerland    [2]LatticeFlow

`{ymerkli,pavol,petar}@latticeflow.ai`
`martin.vechev@inf.ethz.ch`

## Abstract

While current methods for training robust deep learning models optimize robust accuracy, they significantly reduce natural accuracy, hindering their adoption in practice. Further, the resulting models are often both robust and inaccurate on numerous samples, providing a false sense of safety for those. In this work, we extend prior works in three main directions. First, we explicitly train the models to jointly maximize robust accuracy and minimize robust inaccuracy. Second, since the resulting models are trained to be robust only if they are accurate, we leverage robustness as a principled abstain mechanism. Finally, this abstain mechanism allows us to combine models in a compositional architecture that significantly boosts overall robustness without sacrificing accuracy. We demonstrate the effectiveness of our approach for empirical robustness on four recent state-of-the-art models and four datasets. For example, on `CIFAR-10` with $\varepsilon_\infty = 1/255$, we successfully enhanced the robust accuracy of a pre-trained model from 26.2% to 87.8% while even slightly increasing its natural accuracy from 97.8% to 98.0%.

## 1   Introduction

Despite significant progress in training robust models [3, 7, 16, 23, 30], there are two key limitations that hinder the wider adoption of robust models in practice.

**Existing Models are Robustly Inaccurate**   First, existing works usually only report robust accuracy, i.e., samples for which the model robustly predicts the correct label. Meanwhile, the issue of robust inaccuracy, i.e., samples that are robustly misclassified with a wrong label, is usually not even reported (see Section 3). This is especially problematic for safety-critical models, where the robustness can be mistakenly used as a safety argument. We quantify the severity of this issue in Table 1, by evaluating recent state-of-the-art robust models. As can be seen, recent models contain up to 15% of robust inaccurate samples and the ratio of such samples worsens with smaller perturbation regions.

Second, existing robust training methods improve the model robustness, but they also typically degrade the standard accuracy on unperturbed inputs. To address this limitation, a number of recent works study this issue in detail and propose new methods to mitigate it [24, 26, 28, 33].

Table 1: Percentage of robust and inaccurate samples for various recent robust models (cf. Section 6).

| | CIFAR-10 | | | | | CIFAR-100 |
|---|---|---|---|---|---|---|
| | Zhang et al. [34] | Carmon et al. [4] | Gowal et al. [18] | Ding et al. [10] | Wang et al. [31] | Rebuffi et al. [27] |
| $\mathcal{B}^\infty_{1/255}$ | 4.6% | 3.6% | 2.9% | 5.32% | 2.74% | 15.2% |
| $\mathcal{B}^\infty_{4/255}$ | 3.3% | 1.1% | 0.9% | 1.37% | 0.84% | 4.3% |
| $\mathcal{B}^\infty_{8/255}$ | 2.6% | 0.8% | 1.3% | 1.48% | 1.01% | 3.9% |

2022 Trustworthy and Socially Responsible Machine Learning (TSRML 2022) co-located with NeurIPS 2022.

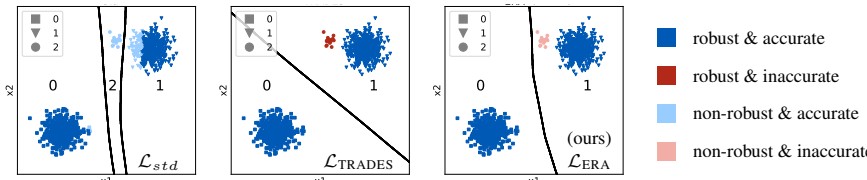

Figure 1: Decision regions for models trained via standard training $\mathcal{L}_{std}$, adversarial training $\mathcal{L}_{\text{TRADES}}$ [34], and our training $\mathcal{L}_{\text{ERA}}$ (Equation 4). Here, our $\mathcal{L}_{\text{ERA}}$ achieves the same robust accuracy as $\mathcal{L}_{\text{TRADES}}$ but avoids all robust inaccurate samples by making them non-robust. Note that all models predict over all three classes, however, the decision regions for class 2 of the $\mathcal{L}_{\text{TRADES}}$ and $\mathcal{L}_{\text{ERA}}$ trained models are too small to be visible. For more details, please refer to Appendix A.2.

**Our Work** In this work, we advance the line of work that aims to boost robustness without sacrificing accuracy, but we approach the problem from a new perspective – by avoiding robust inaccuracy.

Concretely, we propose a new training method that jointly maximizes robust accuracy while minimizing robust inaccuracy. We illustrate the effect of our training on a synthetic dataset (three classes sampled from Gaussian distributions) in Figure 1, showing the decision boundaries of three models, trained using standard training $\mathcal{L}_{std}$, adversarial training $\mathcal{L}_{\text{TRADES}}$ [34], and our training $\mathcal{L}_{\text{ERA}}$ (Equation 4). First, observe that while the $\mathcal{L}_{std}$ trained model achieves $100\%$ accuracy, only $91.1\%$ of these samples are robust (and accurate). When using $\mathcal{L}_{\text{TRADES}}$, we can observe the robustness vs accuracy tradeoff – the robust accuracy improves to $98.4\%$ at the expense of $1.6\%$ (robust) inaccuracy. In contrast, using $\mathcal{L}_{\text{ERA}}$, we retain the high robust accuracy of $98.4\%$ but avoid all robust inaccurate samples by appropriately shifting the decision boundary, rendering them non-robust.

Since our models are trained to be robust only if they are accurate, we leverage robustness as a principled abstain mechanism. This abstain mechanism then allows us to combine models in a compositional architecture that significantly boosts overall robustness without sacrificing accuracy. Concretely, in Figure 1, we would define a selector model that abstains on all non-robust samples. Then, the abstained (non-robust) samples are evaluated by the standard trained model $\mathcal{L}_{std}$, while the selected samples are evaluated using the robust model $\mathcal{L}_{\text{ERA}}$. This allows us to achieve the best of both models – high robust accuracy ($98.4\%$), high natural accuracy ($100\%$), and no robust inaccuracy. We release our code at: `https://github.com/ymerkli/robust-abstain`.

## 2 Related Work

Several recent works investigate the robustness and accuracy tradeoff both theoretically [11, 33] and practically. For example, Stutz et al. [28] considers a new method based on on-manifold adversarial examples, which are more aligned with the true data distribution than the $\ell_p$-noise models. Mueller et al. [24] focuses on deterministic certification and proposes using compositional models to control the robustness and accuracy tradeoff. In our work, we also use compositional models, but focus on empirical robustness. Our selector formulation is based on a new training that minimizes robust inaccuracy and can be used to fine-tune any existing robust model. Further, we provide individual robustness at inference time, rather than distributional robustness considered in prior works.

Simultaneously, a growing body of work extends models with an abstain option. Existing approaches include selection mechanisms such as selection function [5, 15, 24], softmax response [14, 29], or explicit abstain class [21, 22]. In our work, we explore an alternative selection mechanism that uses model robustness. The advantage of this formulation is that the selector provides strong guarantees for each sample and never produces false-positive selections. The disadvantage is that it introduces a significant runtime overhead, compared to many other methods that require only a single forward pass.

Finally, some recent works also consider learning on misclassified examples. For example, MMA [10] maximizes the margins of correctly classified examples while minimizing the classification loss on misclassified examples. MART [31] combines the standard adversarial risk with a consistency loss that optimizes misclassified examples towards robust predictions. Note, that this formulation actively encourages the model toward robust inaccurate predictions, while our work does the opposite – we minimize robust inaccuracy by penalizing robust misclassified examples.

# 3 Preliminaries

Let $f_\theta \colon \mathbb{R}^d \to \mathbb{R}^k$ be a neural network classifying inputs $\boldsymbol{x} \in \mathcal{X} \subseteq \mathbb{R}^d$ to outputs $\mathbb{R}^k$ (e.g., logits or probabilities). The hard classifier induced by the network is given as $F_\theta(\boldsymbol{x}) = \arg\max_{i \in \mathcal{Y}} f_\theta(\boldsymbol{x})_i$, where $f_\theta(\boldsymbol{x})_i$ is the output for the $i$-th class and $\mathcal{Y}, |\mathcal{Y}| = k$ is the finite set of discrete labels.

**Natural Accuracy**   Given a distribution over input-label pairs $\mathcal{D}$ and a classifier $F_\theta \colon \mathcal{X} \to \mathcal{Y}$, an input-label pair $(\boldsymbol{x}, y)$ is considered accurate iff the classifier $F_\theta$ predicts the correct label $y$ for $\boldsymbol{x}$:

$$\mathcal{R}_{nat}(F_\theta) = \mathbb{E}_{(\boldsymbol{x},y)\sim\mathcal{D}} \quad \mathbf{1}\{F_\theta(\boldsymbol{x}) = y\}$$

**Robust Accuracy**   Given an input-label pair $(\boldsymbol{x}, y)$, we say that the classifier $F_\theta$ is robust and accurate iff it predicts the correct label $y$ for all samples from a predefined region $\mathcal{B}_\varepsilon^p(\boldsymbol{x})$, such as a $\ell_p$-norm ball centered at $\boldsymbol{x}$ with radius $\varepsilon$, i.e., $\mathcal{B}_\varepsilon^p(\boldsymbol{x}) \coloneqq \{\boldsymbol{x}' \colon ||\boldsymbol{x}' - \boldsymbol{x}||_p \leq \varepsilon\}$. Formally:

$$\mathcal{R}_{rob}^{acc}(F_\theta) = \mathbb{E}_{(\boldsymbol{x},y)\sim\mathcal{D}} \quad \mathbf{1}\{F_\theta(\boldsymbol{x}) = y\} \wedge \mathbf{1}\{\forall \boldsymbol{x}' \in \mathcal{B}_\varepsilon^p(\boldsymbol{x}). \; F_\theta(\boldsymbol{x}') = F_\theta(\boldsymbol{x})\} \tag{1}$$

**Robust Inaccuracy**   Similarly to robust accuracy, an input-label pair $(\boldsymbol{x}, y)$ is considered robustly inaccurate iff the classifier $F_\theta$ predicts an incorrect label $F_\theta(\boldsymbol{x}) \neq y$ and $F_\theta$ is robust towards that misprediction for all inputs in $\mathcal{B}_\varepsilon^p(\boldsymbol{x})$. Formally, the robust inaccuracy is defined as:

$$\mathcal{R}_{rob}^{\neg acc}(F_\theta) = \mathbb{E}_{(\boldsymbol{x},y)\sim\mathcal{D}} \quad \mathbf{1}\{F_\theta(\boldsymbol{x}) \neq y\} \wedge \mathbf{1}\{\forall \boldsymbol{x}' \in \mathcal{B}_\varepsilon^p(\boldsymbol{x}). \; F_\theta(\boldsymbol{x}') = F_\theta(\boldsymbol{x})\} \tag{2}$$

# 4 Reducing Robust Inaccuracy

In this section, we present our training method that extends existing robust training approaches by also considering samples that are robust but inaccurate. We start by describing a high-level problem statement which we then instantiate for empirical robustness.

**Problem Statement**   Given a distribution over input-label pairs $\mathcal{D}$, our goal is to find model parameters $\theta$ such that the resulting model maximizes robust accuracy, while at the same time minimizing robust inaccuracy. Concretely, this translates to the following optimization objective:

$$\arg\min_\theta \mathbb{E}_{(\boldsymbol{x},y)\sim\mathcal{D}} \quad \underbrace{\beta \cdot \mathcal{L}_{rob}(\boldsymbol{x}, y)}_{\text{optimize robust accuracy}} \quad + \quad \underbrace{\mathbf{1}\{F_\theta(\boldsymbol{x}) \neq y\} \cdot \mathcal{L}_{rob}^{\neg acc}(\boldsymbol{x}, y)}_{\text{penalize robust inaccuracy}} \tag{3}$$

where $\beta \in \mathbb{R}^+$ is a regularization term, $\mathbf{1}\{F_\theta(\boldsymbol{x}) \neq y\}$ is an indicator function denoting samples for which the model is inaccurate, and $\mathcal{L}_{rob}(\boldsymbol{x}, y)$ with $\mathcal{L}_{rob}^{\neg acc}(\boldsymbol{x}, y)$ are loss functions that optimize robust accuracy and penalize robust inaccuracy, respectively. Here, the first loss function $\mathcal{L}_{rob}(\boldsymbol{x}, y)$ is standard and can be directly instantiated using existing approaches. The main challenge comes in defining the second loss term, as well as ensuring that the resulting formulation is easy to optimize, e.g., by defining a smooth approximation of the non-differentiable indicator function.

**Adversarial Training**   We instantiate the loss function from Equation 3 as follows:

$$\mathcal{L}_{\text{ERA}} = \beta \cdot \mathcal{L}_{\text{TRADES}}(f_\theta, (\boldsymbol{x}, y)) + (1 - f_\theta(\boldsymbol{x})_y) \min_{\boldsymbol{x}' \in \mathcal{B}_\varepsilon^p(\boldsymbol{x})} \ell_{\text{CE}}(f_\theta(\boldsymbol{x}'), \arg\max_{c \in \mathcal{Y}\setminus\{F_\theta(\boldsymbol{x})\}} f_\theta(\boldsymbol{x}')_c) \tag{4}$$

Below, we introduce each term in more detail and discuss the motivation behind our formulation.

$\mathcal{L}_{rob}$   To instantiate $\mathcal{L}_{rob}$, we can use any existing adversarial training method [10, 16, 31, 34]. For example, considering TRADES [34], $\mathcal{L}_{rob}$ is instantiated as:

$$\mathcal{L}_{\text{TRADES}} \coloneqq \ell_{\text{CE}}(f_\theta(\boldsymbol{x}), y) + \gamma \max_{\boldsymbol{x}' \in \mathcal{B}_\varepsilon^p(\boldsymbol{x})} D_{\text{KL}}(f_\theta(\boldsymbol{x}), f_\theta(\boldsymbol{x}')) \tag{5}$$

where $D_{\text{KL}}$ is the Kullback-Leibler divergence [20].

$\mathbf{1}\{F_\theta(\boldsymbol{x}) \neq y\}$   Next, we consider the indicator function, which encourages learning on inaccurate samples. Since the indicator function is computationally intractable, we replace the hard qualifier by a soft qualifier $1 - f_\theta(\boldsymbol{x})_y$. The soft qualifier will be small for accurate and large for inaccurate samples, thus providing a smooth approximation of the original indicator function.

$\mathcal{L}_{rob}^{\neg acc}$   Third, we define the loss that penalizes robust but inaccurate samples. This can be formulated similar to the adversarial training objective [23], however, instead of optimizing the prediction of the adversarial example $f_\theta(\boldsymbol{x}')$ towards the correct label $y$, we optimize towards the most likely adversarial label $\arg\max_{c \in \mathcal{Y} \setminus \{F_\theta(\boldsymbol{x})\}} f_\theta(\boldsymbol{x}')_c$. This leads to the following formulation:

$$\min_{\boldsymbol{x}' \in \mathcal{B}_\varepsilon^p(\boldsymbol{x})} \ell_{\text{CE}}(f_\theta(\boldsymbol{x}'), \arg\max_{c \in \mathcal{Y} \setminus \{F_\theta(\boldsymbol{x})\}} f_\theta(\boldsymbol{x}')_c) \tag{6}$$

The purpose of the $\mathcal{L}_{rob}^{\neg acc}$ is to penalize robustness by making the model non-robust. As a result, it is sufficient to consider only a single non-robust example, thus the minimization in the loss objective.

## 5   Boosting Robustness without Accuracy Loss

Next, we extend the models trained so far by leveraging robustness as a principled abstain mechanism.

**Abstain Model**   Given input space $\mathcal{X} \subseteq \mathbb{R}^d$ and label space $\mathcal{Y}$, a model with an abstain option [12] is a pair of functions $(F_\theta, S)$, where $F_\theta \colon \mathcal{X} \to \mathcal{Y}$ is a classifier and $S \colon \mathcal{X} \to \{0, 1\}$ is a selector, acting as a binary qualifier for $F_\theta$. Let $S(\boldsymbol{x}) = 0$ indicate that the model abstains on input $\boldsymbol{x} \in \mathcal{X}$, while $S(\boldsymbol{x}) = 1$ indicates that the model commits to the classifier $F_\theta$ and predicts $F_\theta(\boldsymbol{x})$. In our case, we consider a robustness indicator selector $S_{\text{RI}}$, which abstains on all non-robust samples:

$$S_{\text{RI}}(\boldsymbol{x}) = \mathbf{1}\{\forall \boldsymbol{x}' \in \mathcal{B}(\boldsymbol{x}) \colon F_\theta(\boldsymbol{x}') = F_\theta(\boldsymbol{x})\} \tag{7}$$

**Robustness Guarantees: Robust Selection**   Similar to robust accuracy, the robustness of an abstain model needs to be evaluated with respect to a threat model. In our work, we consider the same threat model as for the underlying model $F_\theta$, namely $\mathcal{B}_\varepsilon^p(\boldsymbol{x}) \coloneqq \{\boldsymbol{x}' \colon ||\boldsymbol{x}' - \boldsymbol{x}||_p \leq \varepsilon\}$, a $\ell_p$-norm ball centered at $\boldsymbol{x}$ with radius $\varepsilon$. Then, we define the robust selection of an abstain model as follows:

$$\mathcal{R}_{rob}^{sel}(S) = \mathbb{E}_{(\boldsymbol{x}, y) \sim \mathcal{D}} \quad \mathbf{1}\{\forall \boldsymbol{x}' \in \mathcal{B}_\varepsilon^p(\boldsymbol{x}). \, S(\boldsymbol{x}') = 1\}$$

That is, we say that a model robustly selects $\boldsymbol{x}$ if the selector $S$ would select all valid perturbations $\boldsymbol{x}' \in \mathcal{B}_\varepsilon^p(\boldsymbol{x})$. Combined with our definition of $S_{\text{RI}}$, we obtain the following criterion (cf. Appendix A.3):

$$\mathcal{R}_{rob}^{sel}(S_{\text{RI}}) = \mathbb{E}_{(\boldsymbol{x}, y) \sim \mathcal{D}} \mathbf{1}\{\forall \boldsymbol{x}' \in \mathcal{B}_{2 \cdot \varepsilon}^p(\boldsymbol{x}). \, F_\theta(\boldsymbol{x}') = F_\theta(\boldsymbol{x})\}$$

In other words, to guarantee that the selector $S_{\text{RI}}$ is robust for all $\boldsymbol{x}' \in \mathcal{B}_\varepsilon^p(\boldsymbol{x})$, we in fact need to check robustness of the model $F_\theta$ to double that region $\boldsymbol{x}' \in \mathcal{B}_{2 \cdot \varepsilon}^p(\boldsymbol{x})$. This is important in order to obtain the correct guarantees and is reflected in our evaluation in Section 6.

**Compositional Architectures**   Consider abstain model $(F_\theta, S)$ and dataset $\mathcal{D}$. Selector $S$ partitions $\mathcal{D}$ into two disjoint subsets – the abstained inputs $\mathcal{D}_{\neg s}$ and the selected inputs $\mathcal{D}_s$ for which $F_\theta$ makes a prediction. Depending on the task, making a best-effort prediction on all samples $\mathcal{D}_s \cup \mathcal{D}_{\neg s}$ may be desirable, which leads to compositional architectures, already used by prior works [24, 32].

Let $H = ((F_{robust}, S), F_{core})$ be a 2-compositional architecture consisting of a selection mechanism $S$, a robustly trained model $F_{robust}$, and a core model $F_{core}$. Given an input $\boldsymbol{x} \in \mathcal{X}$, the selector $S$ decides whether the model is confident on $\boldsymbol{x}$ and commits to the robust model $F_{robust}$ or whether the model should abstain and fall back to the core model $F_{core}$. Formally:

$$H(\boldsymbol{x}) = S(\boldsymbol{x}) \cdot F_{robust}(\boldsymbol{x}) + (1 - S(\boldsymbol{x})) \cdot F_{core}(\boldsymbol{x}) \tag{8}$$

While $F_{robust}, F_{core}$ can be chosen arbitrarily, we here combine robust trained models (which have lower natural accuracy), with standard trained models (which have high natural accuracy but low robustness). The performance of $H$ then depends on the quality of the selector $S$.

## 6   Evaluation

We evaluate our approach on four different datasets, four recent state-of-the-art empirically robust models, including selected top models from RobustBench [8]. We show the following key results:

- Fine-tuning models with our proposed loss successfully decreases robust inaccuracy and provides a Pareto front of models with different robustness tradeoffs.
- Our 2-compositional models significantly improve robustness by up to $+61\%$ and slightly increase the natural accuracy by up to $+0.2\%$ (for $\mathcal{B}_{1/255}^\infty$ and $\mathcal{B}_{2/255}^\infty$).

We perform all experiments on a single GeForce RTX 3090 GPU and use PyTorch [25] for our implementation. The hyperparameters used for our experiments are provided in Appendix A.2.

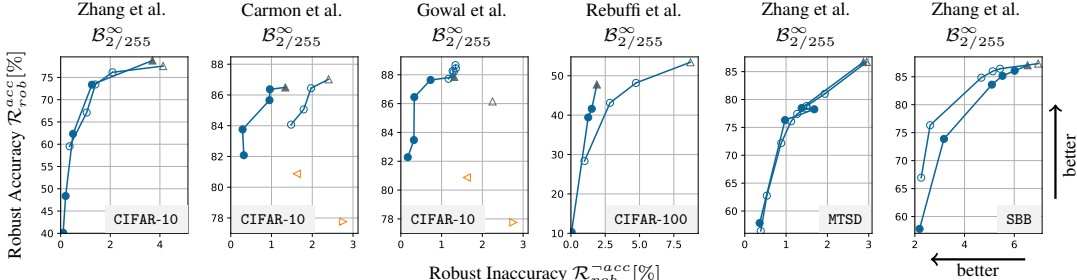

Figure 2: Robust accuracy ($\mathcal{R}_{rob}^{acc}$) and robust inaccuracy ($\mathcal{R}_{rob}^{\neg acc}$) of existing robust models ($\triangle$, $\blacktriangle$, $\triangleleft$, $\triangleright$), and models fine-tuned with our loss ($\circ$, $\bullet$). Our approach consistently reduces robust inaccuracy across various datasets, existing models and different regularization levels $\beta$.

**Models** Our proposed training method requires neither retraining from scratch nor modifications to existing classifiers, thus our approach can be applied to fine-tune a wide range of existing models[1]. To demonstrate this, we use existing robust pre-trained models from Carmon et al. [4], Gowal et al. [18], Rebuffi et al. [27], and Zhang et al. [34], which were all trained for $\varepsilon_\infty = 8/255$, and all but the last model are top models in RobustBench [8]. In our evaluation, we fine-tune each model for 50 epochs for the considered threat model ($\varepsilon_\infty \in \{1/255, 2/255, 4/255\}$), using $\mathcal{L}_{\text{TRADES}}$ [34] and $\mathcal{L}_{\text{ERA}}$ (ours). Further, we also consider models by Ding et al. [10] and Wang et al. [31] as additional baselines.

**Datasets** We evaluate our approach on two academic datasets – CIFAR-10 and CIFAR-100 [19], and two commercial datasets – Mapillary Traffic Sign Dataset (MTSD) [13] and a Rail Defect Dataset kindly provided by Swiss Federal Railways (SBB). Consider Appendix A.1 for full details.

When training on the CIFAR-10 and CIFAR-100 datasets, we use the AutoAugment (AA) policy by Cubuk et al. [9] as the image augmentation. For the MTSD and SBB datasets, we use standard image augmentations (SA) consisting of random cropping, color jitter, and random translation and rotation. For completeness, our evaluation also includes models trained without any data augmentations.

**Metrics** We use the natural accuracy, robust accuracy, and robust inaccuracy as our main evaluation metrics, as defined in Section 3, but evaluated on the corresponding test dataset. Further, for a fair evaluation, we use 10-step PGD [23] attack during training and strong 40-step APGD$_{\text{CE}}$ [7] for testing.

## 6.1 Reducing Robust Inaccuracy

We first summarize the main results obtained by using our proposed loss function $\mathcal{L}_{\text{ERA}}$. The results in Figure 2 show the robust accuracy ($\mathcal{R}_{rob}^{acc}$) and robust inaccuracy ($\mathcal{R}_{rob}^{\neg acc}$) of different existing robust models fine-tuned via TRADES [34] with ($\blacktriangle$) and without ($\triangle$) data augmentations, and the same models fine-tuned via our $\mathcal{L}_{\text{ERA}}$ with ($\bullet$) and without ($\circ$) data augmentations. Further, we also show MART [31] ($\triangleleft$), and MMA [10] ($\triangleright$) finetuned models as an additional baseline. We can see that our approach improves over the existing models across all datasets. For example, for CIFAR-10 and $\mathcal{B}_{2/255}^\infty$, the Carmon et al. [4] model achieves 86.5% robust accuracy, but also 1.34% robust inaccuracy. In contrast, using our $\mathcal{L}_{\text{ERA}}$, we can obtain a number of models that reduce robust inaccuracy to 0.29%, while still achieving robustness of 83.8%. Similar results are obtained for other models, perturbation regions, and datasets (cf. Appendix A.7). We observe that our approach achieves consistently lower robust inaccuracy compared to adversarial training. Further, by varying the regularization term $\beta$, we obtain a Pareto front of optimal solutions.

## 6.2 Boosting Robustness without Accuracy Loss

Next, we present our results on using robustness as an abstain mechanism (Section 5) and combining $\mathcal{L}_{\text{ERA}}$ trained models with state-of-the-art standard trained models in a compositional architecture (Section 5). Note that, as discussed in Section 5, when evaluating robustness for $\mathcal{B}_\varepsilon^p$, we in fact need to consider $\mathcal{B}_{2 \cdot \varepsilon}^p$ robustness of the abstain model. We compare the following abstain mechanisms:

---

[1]Our method can also be used to train from scratch, in which case a scheduler for $\beta$ should be introduced.

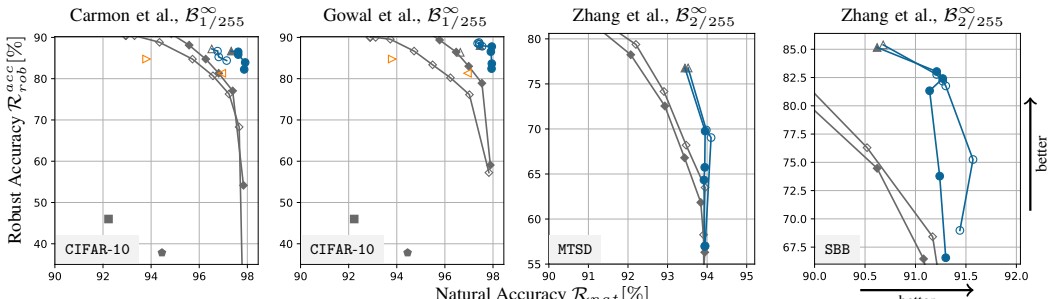

Figure 3: 2-compositional natural ($\mathcal{R}_{nat}$) and robust accuracy ($\mathcal{R}_{rob}^{acc}$) for ERA$_{\text{RI}}$ (○, ●), TRADES$_{\text{RI}}$ (△, ▲), MART$_{\text{RI}}$ (◀), MMA$_{\text{RI}}$ (▷), ACE-COLT$_{\text{SN}}$, ACE-IBP$_{\text{SN}}$ (■, ⬟), and TRADES$_{\text{SR}}$ (◇, ◆) models. The core models used in the compositional architectures are listed in Appendix A.9. We can see that the Pareto front of our method strictly improves over the prior work in the most important region – significantly improving model robustness while the model accuracy does not decrease.

*Softmax Response (SR)* [14], which abstains if the maximum softmax output of the model $f_\theta$ is below a threshold $\tau$ for some input $x' \in \mathcal{B}_\varepsilon^p(x)$, that is:

$$S_{\text{SR}}(x) = \mathbf{1}\{\forall x' \in \mathcal{B}_\varepsilon^p(x)\colon \max_{c \in \mathcal{Y}} f_\theta(x')_c \geq \tau\} \tag{9}$$

Similar to $S_{\text{RI}}$, to guarantee robustness of $S_{\text{SR}}$, we need to check the maximum softmax output of $f_\theta$ on double the region $\mathcal{B}_{2 \cdot \varepsilon}^p(x)$. To evaluate robustness of $S_{\text{SR}}$, we use a modified version of APGD called APGDconf (Appendix A.5). For each considered model (e.g., Carmon et al. [4]), we evaluate its corresponding abstain selector: (◇, ◆) CARMON$_{\text{SR}}$, GOWAL$_{\text{SR}}$, etc. (all fine-tuned using TRADES).

*Robustness Indicator (RI) (our work)*, which abstains if the model $F_\theta$ is non-robust:

$$S_{\text{RI}}(x) = \mathbf{1}\{\forall x' \in \mathcal{B}_\varepsilon^p(x)\colon F_\theta(x') = F_\theta(x)\} \tag{10}$$

Note that, unlike other selectors, our robustness indicator is by design robust against an adversary using the same threat model. For each base model, we consider two instantiations (△, ▲) TRADES$_{\text{RI}}$, and (○, ●) ERA$_{\text{RI}}$ (Equation 4). Further, for CIFAR-10, we also instantiate models from [10, 31] with robustness indicator abstain: MART$_{\text{RI}}$ (◀), and MMA$_{\text{RI}}$ (▷).

*Selection Network (SN)*, which trains a separate neural network $s_\theta\colon \mathcal{X} \to \mathbb{R}$ and selects if:

$$S_{\text{SN}}(x) = \mathbf{1}\{s_\theta(x) \geq \tau\} \tag{11}$$

When evaluating the robustness of an abstain model $(F_\theta, S_{\text{SN}})$, the robustness of both the classifier and the selection network have to be considered. We compare against two instantiations of this approach, both trained using certified training: (■) ACE-COLT$_{\text{SN}}$ [2, 24], and (⬟) ACE-IBP$_{\text{SN}}$ [17, 24].

A summary of the results is shown in Figure 3. Observe that the 2-compositional architectures that use models trained by our method (○, ●) improve over existing methods that optimize robust accuracy (△, ▲, ◀, ▷), as well as over models using softmax response (◇, ◆) or selection network (■, ⬟) to abstain. For example, for CIFAR-10 with $\varepsilon_\infty = {}^1/_{255}$ and the Carmon et al. [4] model, we improve natural accuracy by +0.58% and +0.62%, while decreasing the robustness only by -2.75% and -2.82%, when training with and without data augmentations respectively.

More importantly, our approach significantly improves robustness of highly accurate non-compositional models, with minimal loss of accuracy, which we have summarized in Table 2. We provide full results, including additional models and perturbation bounds in Appendix A.8, and an evaluation of the considered highly accurate non-compositional models in Appendix A.9.

Table 2: Improvement of applying our approach to models trained to optimize natural accuracy only. Here, $\mathcal{R}_{rob}^{acc}$ denotes the robust accuracy and $\mathcal{R}_{nat}$ denotes the standard (non-adversarial) accuracy.

|  | CIFAR-10 Zhao et al. [36], $\mathcal{B}_{1/255}^\infty$ | | CIFAR-100 (WideResNet-28-10), $\mathcal{B}_{2/255}^\infty$ | | MTSD (ResNet-50), $\mathcal{B}_{2/255}^\infty$ | | SBB (ResNet-50), $\mathcal{B}_{2/255}^\infty$ | |
|---|---|---|---|---|---|---|---|---|
| $\mathcal{R}_{rob}^{acc}$ | 26.2 | $\xrightarrow{\textbf{+61.6\%}}$ 87.8 | 3.1 | $\xrightarrow{\textbf{+38.8\%}}$ 41.9 | 40.7 | $\xrightarrow{\textbf{+29.2\%}}$ 69.9 | 44.7 | $\xrightarrow{\textbf{+37.7\%}}$ 82.4 |
| $\mathcal{R}_{nat}$ | 97.8 | $\xrightarrow{\textbf{+0.2\%}}$ 98.0 | 80.17 | $\xrightarrow{\textbf{+0.01\%}}$ 80.18 | 93.8 | $\xrightarrow{\textbf{+0.2\%}}$ 94.0 | 91.4 | $\xrightarrow{\textbf{−0.1\%}}$ 91.3 |

# 7 Conclusion

In this work, we address the robustness vs accuracy tradeoff by avoiding robust inaccuracy and leveraging model robustness as an abstain mechanism. We present a new training method that jointly minimizes robust inaccuracy and maximizes robust accuracy, and that can be instantiated using various existing robust training methods. We show the practical benefits of our approach by leveraging compositional architectures to improve robustness without sacrificing accuracy.

However, there are also limitations and extensions to consider in the future. First, while there are cases where our training improves robust accuracy and reduces robust inaccuracy, it does typically result in a trade-off between the two. An interesting future work is exploring this trade-off further and developing new techniques to mitigate it. Second, given that we compute a Pareto front of optimal solutions, another extension is to consider model cascades that consist of different models along this Pareto front, and progressively fall back to models with higher robust accuracy but also higher robust inaccuracy. Third, we observed that the training becomes much harder as robust inaccuracy approaches zero (i.e., the best case). This is because these remaining robust inaccurate examples are the hardest to fix, and because there are only a few. In our work, we explored using data augmentation to address this issue, but more work is needed to make the training efficient in such a low data regime. Finally, we only consider empirical robustness in our work. Thus, a natural extension is instantiating our problem statement from Equation 3 for deterministic or probabilistic certified robustness.

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

# A Appendix

## A.1 Datasets

We ran our evaluations on four different datasets, namely on `CIFAR-10` and `CIFAR-100` [19], the Mapillary Traffic Sign Dataset (`MTSD`) [13], and a rail defect dataset provided by Swiss Federal Railways (SBB). Additionally, we used a synthetic dataset consisting of two-dimensional data points. In the following, we explain the necessary preprocessing steps to create the publicly available `MTSD` dataset.

**Mapillary Traffic Sign Dataset (MTSD)**   The Mapillary traffic sign dataset [13] is a large-scale vision dataset that includes 52'000 fully annotated street-level images from all around the world. The dataset covers 400 known and other unknown traffic signs, resulting in over 255'000 traffic signs in total. Each street-level image is manually annotated and includes ground truth bounding boxes that locate each traffic sign in the image, as shown in Figure 4a. Further, each ground truth traffic sign annotation includes additional attributes such as ambiguousness or occlusion. Since the focus of this work is on classification, we convert the base `MTSD` dataset to a classification dataset (described below) by cropping to each ground truth bounding box. We show samples from the resulting cropped `MTSD` dataset in Figure 4b.

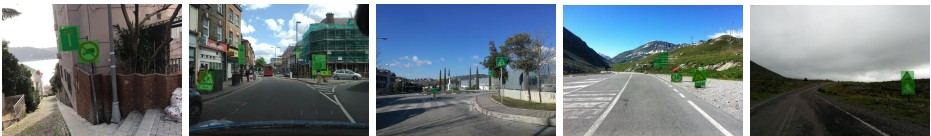

(a) Base Mapillary Traffic Sign Dataset (`MTSD`). The ground truth bounding boxes are visualized in green.

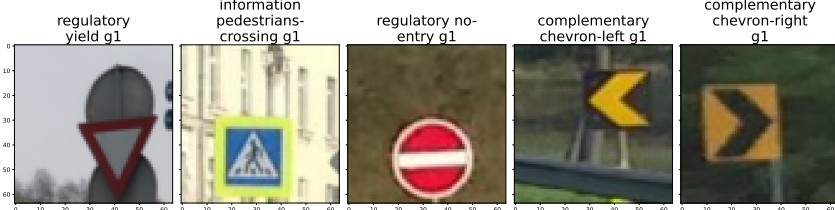

(b) Preprocessed Mapillary Traffic Sign Dataset (`MTSD`).

Figure 4: Illustration of Mapillary Traffic Sign Dataset (`MTSD`) samples. The base dataset consists of street-level images that include annotated ground truth bounding boxes locating the traffic signs (a). We convert the dataset to a classification task by cropping to the ground truth bounding boxes (b).

We convert the `MTSD` objection detection dataset into a classification dataset as follows:

1. Ignore all bounding boxes that are annotated as occluded (sign partly occluded), out-of-frame (sign cut off by image border), exterior (sign includes other signs), ambiguous (sign is not classifiable at all), included (sign is part of another bigger sign), dummy (looks like a sign but is not) [13]. Further, we ignore signs of class *other-sign*, since this is a general class that includes any traffic sign with a label not within the `MTSD` taxonomy.

2. Crop to all remaining bounding boxes and produce a labeled image classification dataset. Cropping is done with slack, i.e. we crop to a randomly upsized version of the original bounding box. Given a bounding box $BB = ([x_{min}, x_{max}], [y_{min}, y_{max}])$, the corresponding upsized bounding box is given as

$$UBB = \big([x_{min} - \lambda\alpha_x(x_{max} - x_{min}), \ x_{max} + \lambda(1 - \alpha_x)(x_{max} - x_{min})],$$
$$[y_{min} - \lambda\alpha_y(y_{max} - y_{min}), \ y_{max} + \lambda(1 - \alpha_y)(y_{max} - y_{min})]\big) \tag{12}$$

where $\alpha_x, \alpha_y \sim \mathcal{U}_{[0,1]}$ [2] and $\lambda$ is the slack parameter, which we set to $\lambda = 1.0$.

3. Resize cropped traffic signs to $(64, 64)$.

---

[2] $\mathcal{U}_{[a,b]}$ is the uniform distribution over the interval $[a, b]$.

**Rail Defect Dataset (SBB)**    The rail defect dataset (SBB) is a proprietary vision dataset collected and annotated by Swiss Federal Railways. It includes images of rails, each of which is annotated with ground truth bounding boxes for various types of rail defects. We note that all the models used in our work for this dataset are trained by the authors and not provided by SBB. In fact, for our work, we even consider a different type of task – classification instead of the original object detection. As a consequence, the accuracy and robustness results presented in our work are by no means representative of the actual models used by SBB.

## A.2 Hyperparameters

**TRADES**    We use $\mathcal{L}_{\text{TRADES}}$ [34] to both train models from scratch and fine-tune existing models. When training models from scratch, we train for 100 epochs using $\mathcal{L}_{\text{TRADES}}$, with an initial learning rate 1e-1, which we reduce to 1e-2 and 1e-3, once 75% and 90% of the total epochs are completed. When fine-tuning models, we train for 50 epochs using $\mathcal{L}_{\text{TRADES}}$, with an initial learning rate 1e-3, which we reduce to 1e-4 once 75% of the total epochs are completed. We use batch size 200, use 10-step PGD [23] to generate adversarial examples during training, and set the $\beta$ parameter in $\mathcal{L}_{\text{TRADES}}$ to $\beta_{TRADES} = 6.0$.

**Robustness Abstain Training**    We fine-tune for 50 epochs using $\mathcal{L}_{\text{ERA}}$ (Equation 4), with an initial learning rate 1e-3, which we reduce to 1e-4 once 75% of the total epochs are completed. We use batch size 200, use 10-step PGD [23] to generate adversarial examples during training, and set $\beta_{TRADES} = 6.0$ for the loss term $\mathcal{L}_{rob} = \mathcal{L}_{\text{TRADES}}$.

**MMA**    We use `MMA` [10] as an additional baseline to compare our models against. In our evaluations, we use the $d_{max} = {}^{12}/_{255}$ trained WideResNet-28-10 published by [10], and fine-tune it using `MMA` with $d_{max} = {}^{4}/_{255}$ for 50 epochs. We decided to fine-tune with $d_{max} = {}^{4}/_{255}$, since we typically evaluate smaller perturbation regions ($\varepsilon_{\infty} \in \{{}^{1}/_{255}, {}^{2}/_{255}, {}^{4}/_{255}\}$), and since Ding et al. [10] claim that $d_{max}$ should usually be set larger than $\varepsilon_{\infty}$ in standard adversarial training. We fine-tune for 50 epochs, using the same hyperparameters as Ding et al. [10], and without using data augmentations.

**MART**    We use `MART` [31] as an additional baseline to compare our models against. In our evaluations, we use the $\varepsilon_{\infty} = {}^{8}/_{255}$ trained WideResNet-28-10 (trained with 500K unlabeled data) published by Wang et al. [31], and fine-tune it using `MART` for the respective smaller perturbation region ($\varepsilon_{\infty} \in \{{}^{1}/_{255}, {}^{2}/_{255}, {}^{4}/_{255}\}$). We fine-tune for 50 epochs, using the same hyperparameters as Wang et al. [31], and without using data augmentations.

**Synthetic Dataset**    In Figure 1, we illustrate the effect of our training on a synthetic three-class dataset, where each class follows a Gaussian distribution. We then use a simple four-layer neural network with 64 neurons per layer, and train it on $N = 1000$ synthetic samples, using $\mathcal{L}_{std}$, $\mathcal{L}_{\text{TRADES}}$ [34], and $\mathcal{L}_{\text{ERA}}$ (Equation 4). For each loss variant, we train for 20 epochs, use a fixed learning rate 1e-1, and batch size 10. For $\mathcal{L}_{\text{TRADES}}$ and $\mathcal{L}_{\text{ERA}}$, we use 10-step PGD [23] to generate adversarial examples during training, and set $\beta_{TRADES} = 6.0$.

Table 3: Robust accuracy ($\mathcal{R}_{rob}^{acc}$) and robust inaccuracy ($\mathcal{R}_{rob}^{\neg acc}$) of the $\mathcal{B}_{2/255}^{\infty}$ $\mathcal{L}_{\text{ERA}}$ ($\beta = 1.0$) finetuned Gowal et al. [18] model, evaluated using both 40-step `APGD` [7] and `AutoAttack` [7].

| | $\mathcal{B}_{1/255}^{\infty}$ | | $\mathcal{B}_{2/255}^{\infty}$ | | $\mathcal{B}_{4/255}^{\infty}$ | | $\mathcal{B}_{8/255}^{\infty}$ | |
|---|---|---|---|---|---|---|---|---|
| | $\mathcal{R}_{rob}^{acc}$ | $\mathcal{R}_{rob}^{\neg acc}$ | $\mathcal{R}_{rob}^{acc}$ | $\mathcal{R}_{rob}^{\neg acc}$ | $\mathcal{R}_{rob}^{acc}$ | $\mathcal{R}_{rob}^{\neg acc}$ | $\mathcal{R}_{rob}^{acc}$ | $\mathcal{R}_{rob}^{\neg acc}$ |
| 40-step `APGD` | 92.93 | 1.01 | 86.45 | 0.33 | 64.09 | 0.06 | 17.20 | 0.0 |
| `AutoAttack` | 92.93 | 1.01 | 86.45 | 0.33 | 63.87 | 0.06 | 16.67 | 0.0 |

### A.3 Robustness Guarantees for Robust Selection

Recall from Section 5 that, given an abstain model $(F_\theta, S)$ and a threat model $\mathcal{B}_\varepsilon^p(\boldsymbol{x}) := \{\boldsymbol{x}' : ||\boldsymbol{x}' - \boldsymbol{x}||_p \leq \varepsilon\}$, $(F_\theta, S)$ is robustly selecting an input $\boldsymbol{x}$ if the selector $S$ selects all valid perturbations $\boldsymbol{x}' \in \mathcal{B}_\varepsilon^p(\boldsymbol{x})$:

$$\mathcal{R}_{rob}^{sel}(S) = \mathbb{E}_{(\boldsymbol{x},y)\sim\mathcal{D}} \quad \mathbf{1}\{\forall \boldsymbol{x}' \in \mathcal{B}_\varepsilon^p(\boldsymbol{x}).\ S(\boldsymbol{x}') = 1\}$$

Further, recall that when evaluating the robustness of an empirical robustness indicator selector $S_{\text{RI}}$ (Equation 7), we in fact need to check robustness of the model $F_\theta$ to double the perturbation region $\boldsymbol{x}' \in \mathcal{B}_{2\cdot\varepsilon}^p(\boldsymbol{x})$, which can be see from the following derivation:

$$
\begin{aligned}
\mathcal{R}_{rob}^{sel}(S_{\text{RI}}) &= \mathbb{E}_{(\boldsymbol{x},y)\sim\mathcal{D}} \quad \mathbf{1}\{\forall \boldsymbol{x}' \in \mathcal{B}_\varepsilon^p(\boldsymbol{x}).\ S_{\text{RI}}(\boldsymbol{x}') = 1\} \\
&= \mathbb{E}_{(\boldsymbol{x},y)\sim\mathcal{D}} \quad \mathbf{1}\{\forall \boldsymbol{x}' \in \mathcal{B}_\varepsilon^p(\boldsymbol{x}).\ \mathbf{1}\{\forall \boldsymbol{x}'' \in \mathcal{B}_\varepsilon^p(\boldsymbol{x}').\ F_\theta(\boldsymbol{x}'') = F_\theta(\boldsymbol{x}')\}\} \\
&= \mathbb{E}_{(\boldsymbol{x},y)\sim\mathcal{D}} \quad \mathbf{1}\{\forall \boldsymbol{x}' \in \mathcal{B}_{2\cdot\varepsilon}^p(\boldsymbol{x}).\ F_\theta(\boldsymbol{x}') = F_\theta(\boldsymbol{x})\}
\end{aligned}
$$

### A.4 Comparing APGD and AutoAttack Robustness

Recall from Section 6 that we use 40-step $\text{APGD}_{\text{CE}}$ [7] (referred to as `APGD`) to evaluate the empirical robustness of classifiers $F_\theta$. `APGD` is one of the adversarial attacks that constitute `AutoAttack` [7], which is an ensemble of adversarial attacks. Concretely, `AutoAttack` consists of $\text{APGD}_{\text{CE}}$ [7], $\text{APGD}^{\text{T}}_{\text{DLR}}$ [7], $\text{FAB}^{\text{T}}$ [6], and `SquareAttack` [1].

In the following, we conduct an ablation study over 40-step `APGD` and `AutoAttack` by comparing the robustness of an $\mathcal{L}_{\text{ERA}}$ trained model. Concretely, we consider the Gowal et al. [18] WideResNet-28-10 model, which was finetuned for $\mathcal{B}_{2/255}^{\infty}$ using our $\mathcal{L}_{\text{ERA}}$ loss (with $\beta = 1.0$) on `CIFAR-10`(cf. Section 6.1). We then evaluate its robust accuracy $\mathcal{R}_{rob}^{acc}$ and robust inaccuracy $\mathcal{R}_{rob}^{\neg acc}$ for the threat models $\varepsilon_\infty \in \{1/255, 2/255, 4/255, 8/255\}$, using both 40-step `APGD` and `AutoAttack`, and show the results in Table 3. Observe that for small perturbation regions $\varepsilon_\infty \in \{1/255, 2/255\}$, the robust accuracy and robust inaccuracy are equivalent for 40-step `APGD` and `AutoAttack`, whereas for larger perturbation regions $\varepsilon_\infty \in \{4/255, 8/255\}$, `AutoAttack` robust accuracy is marginally lower than 40-step `APGD` robust accuracy.

### A.5 Comparing Adversaries for Softmax Response (SR)

Recall from Section 6.2 that we evaluated the robustness of softmax response (SR) abstain models using `APGDconf`, which is a modified version of `APGD` [7] using the alternative adversarial attack objective by [29]. This modified objective optimizes for an adversarial example $\boldsymbol{x}'$ that maximizes the confidence in any label $c \neq F_\theta(\boldsymbol{x})$, instead of minimizing the confidence in the predicted label:

$$\boldsymbol{x}' = \arg\max_{\hat{\boldsymbol{x}}\in\mathcal{B}_\varepsilon^p(\boldsymbol{x})} \max_{c \neq F_\theta(\boldsymbol{x})} f_\theta(\hat{\boldsymbol{x}})_c \tag{13}$$

The resulting adversarial attack finds high confidence adversarial examples, and thus represents an effective attack against a softmax response selector $S_{\text{SR}}$.

In the following, we conduct an ablation study over `APGD` and `APGDconf` by evaluating the robust selection $\mathcal{R}_{rob}^{sel}$ and robust accuracy $\mathcal{R}_{rob}^{acc}$ of an SR abstain model $(F_\theta, S_{\text{SR}})$ using both `APGD` and `APGDconf`. We use the adversarially trained WideResNet-28-10 model by Carmon et al. [4] (taken from RobustBench [8]), trained on `CIFAR-10` for $\varepsilon_\infty = 8/255$ perturbations. We then evaluate the

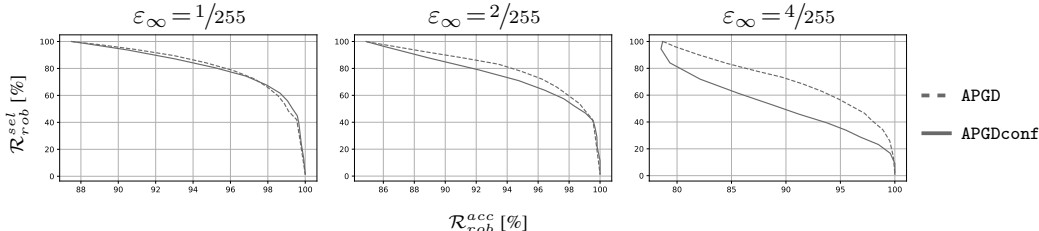

Figure 5: Robust selection ($\mathcal{R}_{rob}^{sel}$) and robust accuracy ($\mathcal{R}_{rob}^{acc}$) for `CIFAR-10` softmax response (SR) abstain models ($F, S_{\text{SR}}$), for varying threshold $\tau \in [0, 1)$ and using the WideResNet-28-10 classifier $F$ by [4]. Each SR abstain model is evaluated via `APGD` [7] and `APGDconf` (Equation 13).

Table 4: Robust selection ($\mathcal{R}_{rob}^{sel}$) and robust accuracy ($\mathcal{R}_{rob}^{acc}$) of empirical robustness indicator abstain models ($F, S_{\text{RI}}$), trained using $\mathcal{L}_{\text{ERA}}$ (Equation 4) and $\mathcal{L}_{\text{DGA}}$ (Equation 14).

| CIFAR-10 | | $\mathcal{B}_{1/255}^{\infty}$ | | $\mathcal{B}_{2/255}^{\infty}$ | |
|---|---|---|---|---|---|
| Pre-trained Model | Finetuning | $\mathcal{R}_{rob}^{sel}$ | $\mathcal{R}_{rob}^{acc}$ | $\mathcal{R}_{rob}^{sel}$ | $\mathcal{R}_{rob}^{acc}$ |
| | $\mathcal{L}_{\text{ERA}}$ | **86.31** | **96.63** | **78.24** | **97.33** |
| Zhang et al. [34] | $\mathcal{L}_{\text{DGA}}$ | 84.98 | 94.92 | 75.73 | 96.22 |
| (ResNet-50) | $\mathcal{L}_{\text{ERA}} + AA$ | **83.44** | **97.47** | **74.63** | **98.31** |
| | $\mathcal{L}_{\text{DGA}} + AA$ | 80.72 | 96.56 | 73.59 | 97.88 |

classifier as an SR abstain model ($F_\theta, S_{\text{SR}}$) with varying threshold $\tau \in [0, 1)$, and report the robust selection and robust accuracy for varying $\ell_\infty$ perturbations in Figure 5. Observe that for small perturbations such as $\varepsilon_\infty = 1/255$, `APGD` and `APGDconf` are mostly equivalent concerning robust selection and robust accuracy. However, for larger perturbations such as $\varepsilon_\infty = 4/255$, the SR abstain model is significantly less robust to `APGDconf` than to standard `APGD`, showing the importance of choosing a suitable adversarial attack. High confidence adversarial examples are generally more likely to be found for larger perturbations, thus an SR selector is significantly less robust to `APGDconf` than to `APGD` for larger perturbations.

### A.6 Loss Function Ablation Study

Additionally to the $\mathcal{L}_{\text{ERA}}$ loss from Equation 4, we consider an alternative loss formulation for training an empirical robustness indicator abstain model. The formulation is based on the Deep Gamblers loss [22], which considers an abstain model ($F_\theta, S$) with an explicit abstain class $a$ as a selection mechanism. Since we consider robustness indicator selection, we replace the output probability of the abstain class $f_\theta(\boldsymbol{x})_a$ with the output probability of the most likely adversarial label. This corresponds to the probability of a sample being non-robust and thus the probability of abstaining under a robustness indicator selector. Similar to $\mathcal{L}_{\text{ERA}}$, we also add the `TRADES` loss [34] to optimize robust accuracy. The resulting loss is then defined as:

$$\mathcal{L}_{\text{DGA}}(f_\theta, (\boldsymbol{x}, y)) = \beta \cdot \mathcal{L}_{\text{TRADES}}(f_\theta, (\boldsymbol{x}, y)) - \log\left(f_\theta(\boldsymbol{x})_y + \max_{c \in \mathcal{Y} \setminus \{F_\theta(\boldsymbol{x})\}} f_\theta(\boldsymbol{x}')_c\right) \quad (14)$$

We conduct an ablation study over the two loss functions, $\mathcal{L}_{\text{ERA}}$ and $\mathcal{L}_{\text{DGA}}$, for `CIFAR-10` and a $\varepsilon_\infty = 8/255$ `TRADES` [34] trained ResNet-50 model. We fine-tune the model for $\ell_\infty$ perturbations of radii $1/255$ and $2/255$, using both $\mathcal{L}_{\text{ERA}}$ and $\mathcal{L}_{\text{DGA}}$, training for 50 epochs each and setting the regularization parameter $\beta = 1.0$. For each loss variant, we train the base model once without data augmentations and once using the AutoAugment (AA) policy [9].

We show the robust accuracy and the robust selection of the resulting robustness indicator abstain models in Table 4. Observe that for all experiments, $\mathcal{L}_{\text{ERA}}$ trained models achieve consistently higher robust accuracy and higher robust selection, compared to $\mathcal{L}_{\text{DGA}}$ trained models. For instance, when training for $\varepsilon_\infty = 1/255$ perturbations without data augmentations, $\mathcal{L}_{\text{ERA}}$ achieves $+1.71\%$ higher robust accuracy and $+1.33\%$ higher robust selection, compared to $\mathcal{L}_{\text{DGA}}$. Similarly, when training with AutoAugment, $\mathcal{L}_{\text{ERA}}$ achieves $+0.91\%$ higher robust accuracy and $+2.72\%$ higher robust selection. Similar results hold for $\varepsilon_\infty = 2/255$ perturbations.

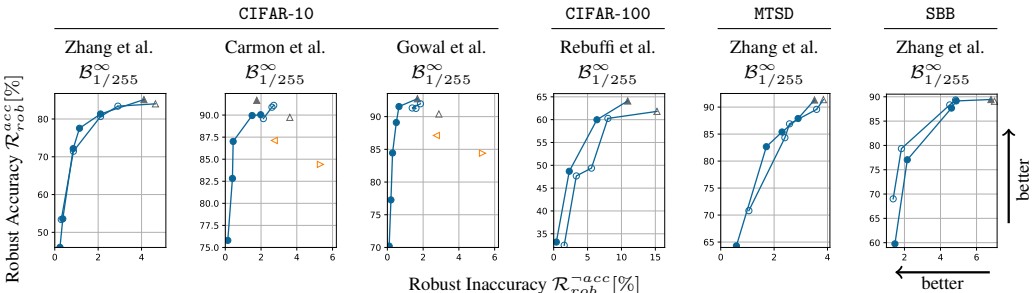

Figure 6: Robust accuracy ($\mathcal{R}_{rob}^{acc}$) and robust inaccuracy ($\mathcal{R}_{rob}^{\neg acc}$) of existing robust models ($\triangle$, $\blacktriangle$) fine-tuned with our proposed loss ($\bigcirc$, $\bullet$). Further, we also show models finetuned via MART [31] ($\triangleleft$) and MMA [10] ($\triangleright$). Our approach consistently reduces the number of robust inaccurate samples across various datasets, existing models and at different regularization levels $\beta$.

## A.7 Additional Experiments on Reducing Robust Inaccuracy

In this section, we present additional experiments on reducing robust inaccuracy.

Similar to the results in Figure 2, we show the robust accuracy ($\mathcal{R}_{rob}^{acc}$) and robust inaccuracy ($\mathcal{R}_{rob}^{\neg acc}$) of different existing models fine-tuned with ($\blacktriangle$) and without ($\triangle$) data augmentations, in Figure 6. At the same time, Figure 6 also shows the same models fine-tuned with our proposed loss with ($\bullet$) and without ($\bigcirc$) data augmentations. We again observe that our approach achieves consistently lower robust robust inaccuracy, compared to existing robust models. For example, on CIFAR-10 and for $\mathcal{B}_{1/255}^{\infty}$, the model from [4] achieves $91.7\%$ robust accuracy but also $1.8\%$ robust inaccuracy. Using our loss $\mathcal{L}_{ERA}$ and varying the regularization term $\beta$, we can obtain a number of models that reduce robust inaccuracy to $0.14\%$ while still achieving robust accuracy of $75.8\%$.

## A.8 Additional Experiments on Boosting Robustness without Accuracy Loss

In this section, we present additional results on combining abstain models with state-of-the-art models trained to achieve high natural accuracy.

Equivalent to Section 6.2, we put the abstain models trained so far in 2-composition (Section 5) with the standard trained core models discussed in Appendix A.9. We show the natural ($\mathcal{R}_{nat}$) and adversarial accuracy ($\mathcal{R}_{rob}^{acc}$) of the resulting 2-compositional architectures in Figure 7.

We again observe that 2-compositional architectures using models trained by our method ($\bigcirc$, $\bullet$) improve over existing methods that solely optimize for robust accuracy ($\triangle$, $\blacktriangle$). Further, our method

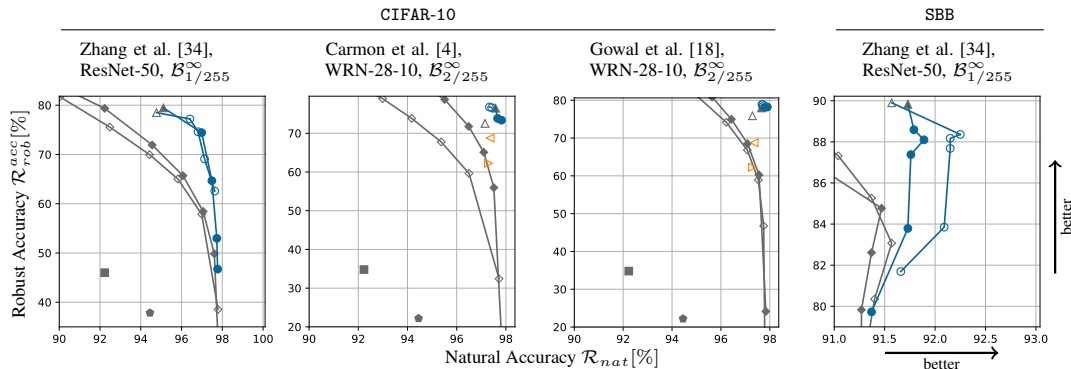

Figure 7: Natural ($\mathcal{R}_{nat}$) and robust accuracy ($\mathcal{R}_{rob}^{acc}$) for 2-compositional ERA$_{RI}$ models ($\bigcirc$, $\bullet$), and 2-compositional TRADES$_{RI}$ ($\triangle$, $\blacktriangle$), MART$_{RI}$ ($\triangleleft$), and MMA$_{RI}$ ($\triangleright$) models. Further, we also consider 2-compositional ACE-COLT$_{SN}$, ACE-IBP$_{SN}$ ($\blacksquare$, $\pentagon$), and 2-compositional TRADES$_{SR}$ ($\diamond$, $\blacklozenge$) models. The core models used in the compositional architectures are listed in Appendix A.9.

mostly improves both the natural and robust accuracy, compared to 2-compositional architectures using softmax response (◇, ◆) or selection network (■, ◐) to abstain. For example, on SBB and the Zhang et al. [34] model at $\varepsilon_\infty = 1/255$, our approach (◯) improves natural accuracy by +0.68%, while decreasing the robust accuracy by only -1.54%.

Further, we show that 2-compositional architectures using models trained by our method achieve significantly higher robustness and mostly equivalent overall accuracy, compared to state-of-the-art non-compositional models trained for high natural accuracy. In Table 5, we show the natural ($\mathcal{R}_{nat}$) and adversarial accuracy ($\mathcal{R}_{rob}^{acc}$) of our 2-compositional models and illustrate the accuracy improvement over the standard trained models discussed in Appendix A.9. For instance, consider CIFAR-10 at $\varepsilon_\infty = 2/255$ and the 2-compositional architecture using the Gowal et al. [18] model as robust model $F_{robust}$. Our model improves the robust accuracy by +75.3% and the natural accuracy by +0.1%, compared to the standard trained model by Zhao et al. [36]. Similar results hold for other models, datasets, and perturbation regions.

Table 5: Improvements of 2-compositional architectures using models $F_{robust}$ trained with our method over non-compositional models trained to optimize natural accuracy only (Appendix A.9).

|  |  | CIFAR-10 | | CIFAR-100 | MTSD | SBB |
|---|---|---|---|---|---|---|
|  | $F_{core}$ | Zhao et al. [36] | | (WideResNet-28-10) | (ResNet-50) | (ResNet-50) |
|  | $F_{robust}$ | Carmon et al. | Gowal et al. | Rebuffi et al. | Zhang et al. | Zhang et al. |
| $\mathcal{B}_{1/255}^\infty$ | $\mathcal{R}_{rob}^{acc}$ | 86.5 (+60.3%) | 87.8 (+61.6%) | 44.0 (+24.1%) | 84.5 (+9.8%) | 88.4 (+12.7%) |
|  | $\mathcal{R}_{nat}$ | 97.6 (-0.2%) | 98.0 (+0.2%) | 80.5 (+0.3%) | 94.1 (+0.3%) | 92.3 (+0.9%) |
| $\mathcal{B}_{2/255}^\infty$ | $\mathcal{R}_{rob}^{acc}$ | 73.4 (+70.5%) | 78.2 (+75.3%) | 41.9 (+38.8%) | 69.9 (+29.2%) | 82.4 (+37.7%) |
|  | $\mathcal{R}_{nat}$ | 97.8 (+0.0%) | 97.9 (+0.1%) | 80.18 (+0.01%) | 94.0 (+0.2%) | 91.3 (-0.1%) |

## A.9  Core Models

Recall from Section 5 that an abstain model $(F, S)$ can be enhanced by a core model $F_{core}$, which makes a prediction on all abstained samples, resulting in 2-compositional architectures. In Section 6.2, we presented an evaluation of 2-compositional architectures, where we used state-of-the-art standard trained models as core models. In Table 6, we show the natural and adversarial accuracy of core models used in Section 6.2, for varying $\ell_\infty$ perturbation regions, where we use 40-step APGD [7] to evaluate robustness.

Table 6: Natural ($\mathcal{R}_{nat}$) and adversarial accuracy ($\mathcal{R}_{rob}^{acc}$) of standard trained core models, used in 2-compositional architectures in Section 6.2 and Appendix A.8.

| Dataset | Model $F_{core}$ | $\mathcal{R}_{nat}$ [%] | $\mathcal{R}_{rob}^{acc}$ [%] | | |
|---|---|---|---|---|---|
|  |  |  | $\mathcal{B}_{1/255}^\infty$ | $\mathcal{B}_{2/255}^\infty$ | $\mathcal{B}_{4/255}^\infty$ |
| CIFAR-10 | Zhao et al. [36] (WideResNet-40-10) | 97.81 | 26.18 | 2.92 | 0.06 |
| CIFAR-100 | (WideResNet-28-10) | 80.17 | 19.9 | 3.06 | 0.15 |
| MTSD | (ResNet-50) | 93.79 | 74.66 | 40.71 | 7.51 |
| SBB | (ResNet-50) | 91.37 | 75.65 | 44.69 | 8.76 |

## A.10  Robustness/Accuracy Dataset Splits

Consider a robustness indicator abstain model $(F_\theta, S_{\texttt{RI}})$ and a labeled dataset $D = \{(\boldsymbol{x}_i, y_i)_{i=1}^N\}$ on which we evaluate the classifier $F_\theta : \mathcal{X} \to \mathcal{Y}$. Based on the robustness and accuracy of the classifier $F_\theta$, we can partition $D$ into four disjoint subsets $D = \{D_{F_\theta}^{r \wedge a}, D_{F_\theta}^{\neg r \wedge a}, D_{F_\theta}^{r \wedge \neg a}, D_{F_\theta}^{\neg r \wedge \neg a}\}$, where:

$$D_{F_\theta}^{r \wedge a} = \{(\boldsymbol{x}, y) \in D : \forall \boldsymbol{x}' \in \mathcal{B}_\varepsilon^p(\boldsymbol{x}). \; F_\theta(\boldsymbol{x}') = F_\theta(\boldsymbol{x}) \wedge F_\theta(\boldsymbol{x}) = y\}$$

$$D_{F_\theta}^{r \wedge \neg a} = \{(\boldsymbol{x}, y) \in D : \forall \boldsymbol{x}' \in \mathcal{B}_\varepsilon^p(\boldsymbol{x}). \; F_\theta(\boldsymbol{x}') = F_\theta(\boldsymbol{x}) \wedge F_\theta(\boldsymbol{x}) \neq y\}$$

$$D_{F_\theta}^{\neg r \wedge a} = \{(\boldsymbol{x}, y) \in D : \exists \boldsymbol{x}' \in \mathcal{B}_\varepsilon^p(\boldsymbol{x}). \; F_\theta(\boldsymbol{x}') \neq F_\theta(\boldsymbol{x}) \wedge F_\theta(\boldsymbol{x}) = y\}$$

$$D_{F_\theta}^{\neg r \wedge \neg a} = \{(\boldsymbol{x}, y) \in D : \exists \boldsymbol{x}' \in \mathcal{B}_\varepsilon^p(\boldsymbol{x}). \; F_\theta(\boldsymbol{x}') \neq F_\theta(\boldsymbol{x}) \wedge F_\theta(\boldsymbol{x}) \neq y\}$$

We illustrate this dataset partitioning on the CIFAR-10 [19] dataset. We consider a TRADES [35] trained ResNet-50 and the WideResNet-28-10 models by Carmon et al. [4], Gowal et al. [18] (taken from Robustbench [8]), where each model is adversarially pretrained for $\varepsilon_\infty = 8/255$ and then fine-tuned via TRADES to the respective $\ell_\infty$ threat model illustrated Table 7. Further, we also consider a standard trained ResNet-50. We then evaluate the robustness and accuracy of each model using 40-step APGD [7]. Considering Table 7, note that standard adversarial training methods do not necessarily eliminate the occurrence of robust inaccurate samples $(\boldsymbol{x}, y) \in D_{F_\theta}^{r \wedge \neg a}$, and that the robust inaccuracy generally increases for smaller perturbation regions. Further, we note that while standard trained models have low robust inaccuracy, they also have low overall robustness, resulting in low overall robust accuracy.

Further, we also illustrate the robustness-accuracy dataset partitioning on CIFAR-100 [19]. We consider a standard trained WideResNet-28-10 and the adversarially trained WideResNet-28-10 by Rebuffi et al. [27]. Again, the model by Rebuffi et al. [27] was pretrained for $\varepsilon_\infty = 8/255$ perturbations and then TRADES fine-tuned for the respective threat model indicated in Table 8. We again evaluate the robustness-accuracy dataset partitioning for varying $\ell_\infty$ perturbations using 40-step APGD [7], and list the exact size of each data split in Table 8.

Notably, we observe that on the model by Rebuffi et al. [27], $15.24\%$ of all test samples are robust but inaccurate for $\varepsilon_\infty = 1/255$ perturbations, which is a significantly larger fraction compared to similar models on CIFAR-10.

Table 7: CIFAR-10 robustness-accuracy dataset partitioning. We consider a TRADES [34] trained ResNet-50, adversarially trained WideResNet-28-10 models [4, 18], and a standard trained ResNet-50. Adversarially trained models are trained for the respective perturbation region. Each model is evaluated for the indicated $\ell_\infty$ threat model, using 40-step APGD [7].

| Threat Model | Data Split | Relative Split Size [%] | | | |
|---|---|---|---|---|---|
| | | Zhang et al. (ResNet-50) | Carmon et al. (WRN-28-10) | Gowal et al. (WRN-28-10) | $\mathcal{L}_{std}$ (ResNet-50) |
| $\mathcal{B}_{1/255}^\infty$ | $\|D_{F_\theta}^{\neg r \wedge \neg a}\|$ | 5.17 | 3.33 | 2.85 | 6.97 |
| | $\|D_{F_\theta}^{r \wedge \neg a}\|$ | 4.64 | 3.61 | 2.88 | 0.0 |
| | $\|D_{F_\theta}^{\neg r \wedge a}\|$ | 6.18 | 3.32 | 3.87 | 74.89 |
| | $\|D_{F_\theta}^{r \wedge a}\|$ | 84.01 | 89.74 | 90.40 | 18.14 |
| $\mathcal{B}_{2/255}^\infty$ | $\|D_{F_\theta}^{\neg r \wedge \neg a}\|$ | 7.94 | 7.38 | 4.86 | 6.97 |
| | $\|D_{F_\theta}^{r \wedge \neg a}\|$ | 4.13 | 2.40 | 2.25 | 0.0 |
| | $\|D_{F_\theta}^{\neg r \wedge a}\|$ | 10.38 | 3.20 | 6.74 | 91.80 |
| | $\|D_{F_\theta}^{r \wedge a}\|$ | 77.55 | 87.02 | 86.15 | 1.23 |
| $\mathcal{B}_{4/255}^\infty$ | $\|D_{F_\theta}^{\neg r \wedge \neg a}\|$ | 13.42 | 8.23 | 6.64 | 6.97 |
| | $\|D_{F_\theta}^{r \wedge \neg a}\|$ | 3.31 | 1.05 | 0.87 | 0.0 |
| | $\|D_{F_\theta}^{\neg r \wedge a}\|$ | 17.19 | 16.87 | 15.96 | 93.03 |
| | $\|D_{F_\theta}^{r \wedge a}\|$ | 66.08 | 73.85 | 76.53 | 0.0 |
| $\mathcal{B}_{8/255}^\infty$ | $\|D_{F_\theta}^{\neg r \wedge \neg a}\|$ | 18.17 | 9.55 | 9.21 | 6.97 |
| | $\|D_{F_\theta}^{r \wedge \neg a}\|$ | 2.64 | 0.76 | 1.31 | 0.0 |
| | $\|D_{F_\theta}^{\neg r \wedge a}\|$ | 29.79 | 27.82 | 23.78 | 93.03 |
| | $\|D_{F_\theta}^{r \wedge a}\|$ | 49.40 | 61.87 | 65.70 | 0.0 |

Table 8: `CIFAR-100` robustness-accuracy dataset partitioning. We consider a standard trained WideResNet-28-10 and the adversarially trained WideResNet-28-10 by Rebuffi et al. [27], trained for the respective perturbation region considered in each evaluation. Each model is evaluated for the indicated $\ell_\infty$ threat model, using 40-step `APGD` [7].

| Threat Model | Data Split | Relative Split Size [%] | |
|---|---|---|---|
| | | Rebuffi et al. (WRN-28-10) | $\mathcal{L}_{std}$ (WRN-28-10) |
| $\mathcal{B}_{1/255}^{\infty}$ | $\|D_{F_\theta}^{\neg r \wedge \neg a}\|$ | 15.20 | 19.80 |
| | $\|D_{F_\theta}^{r \wedge \neg a}\|$ | 15.24 | 0.03 |
| | $\|D_{F_\theta}^{\neg r \wedge a}\|$ | 7.75 | 60.27 |
| | $\|D_{F_\theta}^{r \wedge a}\|$ | 61.81 | 19.9 |
| $\mathcal{B}_{2/255}^{\infty}$ | $\|D_{F_\theta}^{\neg r \wedge \neg a}\|$ | 32.75 | 19.82 |
| | $\|D_{F_\theta}^{r \wedge \neg a}\|$ | 8.71 | 0.01 |
| | $\|D_{F_\theta}^{\neg r \wedge a}\|$ | 5.11 | 77.11 |
| | $\|D_{F_\theta}^{r \wedge a}\|$ | 53.43 | 3.06 |
| $\mathcal{B}_{4/255}^{\infty}$ | $\|D_{F_\theta}^{\neg r \wedge \neg a}\|$ | 30.57 | 19.83 |
| | $\|D_{F_\theta}^{r \wedge \neg a}\|$ | 4.34 | 0.0 |
| | $\|D_{F_\theta}^{\neg r \wedge a}\|$ | 23.16 | 80.02 |
| | $\|D_{F_\theta}^{r \wedge a}\|$ | 41.93 | 0.15 |
| $\mathcal{B}_{8/255}^{\infty}$ | $\|D_{F_\theta}^{\neg r \wedge \neg a}\|$ | 33.70 | 19.83 |
| | $\|D_{F_\theta}^{r \wedge \neg a}\|$ | 3.91 | 0.0 |
| | $\|D_{F_\theta}^{\neg r \wedge a}\|$ | 26.66 | 80.17 |
| | $\|D_{F_\theta}^{r \wedge a}\|$ | 35.73 | 0.0 |

