# OpenReview forum: "Just Avoid Robust Inaccuracy: Boosting Robustness Without Sacrificing Accuracy"
_NeurIPS.cc/2022/Workshop/TSRML — TSRML2022_

### Official Review · Reviewer_dJXL · 2022-10-12

**Overall Rating:** 6

**Summary:**

This paper studies a problem of “robust inaccuracy” where a model is overconfident and makes inaccurate predictions. The paper proposes to reduce “robust accuracy” by an additional loss with adversarial training. The paper also tries to boost robustness without sacrificing accuracy by adding a robust selection.


**Strengths:**

* The “robust inaccuracy” problem is interesting.
* The paper proposes to reduce “robust inaccuracy” and empirically demonstrates the improvement.


**Weaknesses:**

* There are several overclaims:
  - “As can be seen, recent models contain up to 15% of robust inaccurate samples”. In Table 1, most of the models have percentages of robust and inaccurate samples within around 5%. (Only one model has 15.2%, while others are all within 5.32%.) Thus the studied issue is not as severe as claimed.
  - “Robustness Guarantees: Robust Selection”. I don’t see how the proposed selection method has any robustness guarantee. This paper considers empirical robustness and is thus not verified or certified.
* The studied problem does not seem to be significant enough (the studied issue is not very severe in existing models), and the empirical improvement is small (e.g., in Section 6.1, error 1.34% -> 0.29%).
* According to the title “Just Avoid Robust Inaccuracy”, while the focus seems to be on reducing “robust inaccuracy”, Section 5 and Section 6.2 on “Boosting Robustness without Accuracy Loss” does not seem to be coherent with the focus of the paper.


**Overall Recommendation:**

The topic itself is interesting, but not significant enough.

Presentation seems to have some issues as it contains several overclaims and part of the paper does not seem to be coherent with the main focus.

**Review Confidence:**

3: The reviewer is fairly confident that the evaluation is correct

---

### Official Review · Reviewer_kiB6 · 2022-10-20
**Interesting paper with unclear conclusions**

**Overall Rating:** 6

**Summary:**

This paper highlights that robust models are often both robust and inaccurate (i.e., robust inaccuracy) on numerous examples. The authors introduce a method to jointly maximize robust accuracy and minimize robust inaccuracy (used for fine-tuning). They demonstrate that robustness can be used as an abstain mechanism. Experiments show that (for small perturbation radii) their fine-tuning method improves both robust accuracy and natural accuracy.

**Strengths:**

* The paper pinpoints a interesting issue: robust points may be inaccurate.
* The author propose a solution ($\mathcal{L}_{\textrm{ERA}}$)
* Experiments demonstrate that the resulting models obtain higher robust accuracy at similar robust inaccuracy levels (or they can reduce robust inaccuracy at the cost of a smaller robust accuracy).

**Weaknesses:**

* The paper uses small perturbation radii, e.g. $\epsilon = \\{1/255, 2/255, 4/255 \\}$, which makes the comparison with other robust models difficult. By taking a robust model trained against $\epsilon=8/255$ and evaluating it on a smaller $\epsilon$, I would expect the robust inaccuracy to increase (compared to a standard non-robust classifier) as robust models are, in essence, trained to be smooth. I would suggest the authors to train robust models at smaller $\epsilon$ to provide an additional comparison point.

* Despite this possible mismatch, the proposed technique does not strictly improve over pre-trained robust models. Figure 2 shows that in many cases, the robust models are on the Pareto front (only MART and MMA models do worse). Hence, there is a clear trade-off in terms of robust accuracy. Taking this to the extreme, I am curious to see what the robust inaccuracy of the standard model is. Is it lower than the resulting models with large $\beta$? Note that I still consider the ability to trade-off these 2 quantities as being valuable.

* Figure 3 seems to indicate that the abstain mechanism can readily be used on pre-trained models as these models are on the Pareto front. As a result, I am a bit unclear about the link between both contributions (abstaining and ERA).

* The evaluation protocol for the abstain mechanism is unclear. The text simply states: "the robustness of both the classifier and the selection network have to be considered" or "we [...] need to consider $\mathcal{B}_{2\epsilon}^p$". On a similar note, I don't understand Table 5 in the appendix, the table seems to indicate robust accuracy on CIFAR-10 increases by at least 60% at $\epsilon = 1/155$, thereby reaching 86%. If I backtrack this would mean that the original model gets less than 40% robust accuracy against $\epsilon = 1/155$ (which is clearly wrong). Or do the authors mean that their fine-tuned models are significantly worse that the non-finetuned models (thereby making it look like the method is much better than it seems)?

* In general the natural accuracies reported seems quite high (98% on CIFAR-10). Is that because the models were pretrained on additional data?

**Overall Recommendation:**

Overall, I found the paper interesting and found the context of robust inaccuracy intriguing. I feel that the experimental work could be significantly improved and the presentation of the work should be clearer (lots of important details are missing).

**Review Confidence:**

5: The reviewer is absolutely certain that the evaluation is correct and very familiar with the relevant literature

---

### Official Review · Reviewer_L2wS · 2022-10-20
**Review: Just Avoid Robust Inaccuracy: Boosting Robustness Without Sacrificing Accuracy**

**Overall Rating:** 7

**Summary:**

In this paper, the authors propose training NNs to be robust by jointly maximizing robustness of correctly predicted inputs and minimizing the robustness of incorrectly predicted inputs. If a NN has very few inputs that it incorrectly predicts and is robust on, then robustness can be used as a selector function for abstention. Authors use this idea to design a compositional model made of a non-robust model (high standard accuracy) and a robust model (high robust accuracy). Overall, their method achieves high robustness without compromising too much on standard performance, thereby improving usability in practical settings.

**Strengths:**

- The paper is well written and easy to follow. The motivation has been adequately laid out. The proposed idea is conceptually and technically sound. Comparison with related works is properly discussed. I particularly like the use of a toy example to show the expected functionality of the proposed method.

- On two datasets, authors empirically demonstrate that their method improves the accuracy-robustness tradeoff of models pre-trained using existing methods. Since there method is a fine-tuning one, it can be used on top of any pre-trained model to improve its accuracy/robustness tradefoff. This make their method general-purpose, greatly improving its usability.


**Weaknesses:**

- Need to deploy two models instead of one, which can be a major issue when deploying in resource-restricted environments.

- Consider an input that is incorrectly classified by the robust model. Based on the design of the proposed defense, such an input will be redirected to the core model. This can be a possible attack vector for an adversary, i.e., jointly target misclassification on the robust and standard model. Such an adaptive attack has not been discussed in the paper.

**Overall Recommendation:**

Overall, I believe this paper proposed a promising direction towards training empricially robust models to have higher natural/standard accuracy. I believe this paper has the potential to bring to forefront the discussion regarding how robust inaccuracies need to be handled to benefit the overall robustness posture of the model. Therefore, I am leaning towards acceptance.


**Review Confidence:**

3: The reviewer is fairly confident that the evaluation is correct

---

### Decision · Program_Chairs · 2022-10-23

**Decision:**

Accept

**Comment:**

Following the unanimous recommendations from reviewers, the submission is accepted.